# Beyond Clicking:
# A Step Towards Generalist GUI Grounding via Text Dragging

## Abstract

Graphical user interface (GUI) grounding, the process of mapping human instructions to GUI actions, serves as a fundamental basis to autonomous GUI agents. While existing grounding models achieve promising performance to simulate the mouse click action on various click-based benchmarks, another essential mode of mouse interaction, namely dragging, remains largely underexplored. Yet, dragging the mouse to select and manipulate textual content represents a prevalent and important usage in practical GUI scenarios. To narrow this gap, we first introduce GUI-Drag, a diverse dataset of 161K text dragging examples synthesized through a scalable pipeline. To support systematic and robust evaluation, we further construct ScreenDrag, a benchmark with 5,333 examples spanning three levels of interface context, together with three dedicated metrics designed for assessing text dragging capability. Models trained on GUI-Drag with an efficient continual training strategy achieve substantial improvements on ScreenDrag, while preserving the original click-based performance on ScreenSpot, ScreenSpot-v2, and OSWorld-G. Our work encourages further research on broader GUI grounding beyond just clicking and paves way toward a truly generalist GUI grounding model.

## 1 Introduction

GUI (Graphical user interface) agents based on (multimodal) large language models (LLMs) that can autonomously perceive and act in the digital world have great promise to significantly boost human productivity (Zheng et al., 2024; Qin et al., 2025; OpenAI., 2025). Recent efforts including but are not limited to architecture designs (Wu et al., 2025b; Jing et al., 2025), memory (Gao et al., 2025; Yoran et al., 2024) and grounding (Xie et al., 2025; Gou et al., 2025; Tang et al., 2025) have made significant progress towards this goal. Among these, grounding plays a crucial role by translating the natural language instructions into the actionable operations within the digital world.

The mouse is the primary tool to ground human intent in the digital world, with mouse clicking serving as the dominant mode of engagement. To better understand and simulate the mouse click action, numerous works have focused on both modeling (Lin et al., 2025; Gou et al., 2025; Luo et al., 2025; Lu et al., 2025; Wu et al., 2025a; Liu et al., 2025b) and benchmark development (Li et al., 2025; Wu et al., 2024; Nayak et al., 2025). While the click action is fundamental, it captures only one dimension of mouse interaction. The mouse, by design, supports two complementary modes of operation: discrete clicks, executed through quick taps, and continuous dragging, performed by holding down the button and moving the pointer. A generalist grounding model must therefore encompass the full spectrum of the mouse actions, including dragging. In this study, we identify text dragging as a critical capability with substantial practical value and try to push the boundary of existing grounding models with it.

Text dragging is prevalent and important in daily routine, particularly in professional productivity applications such as Word, PowerPoint, and PDF readers, where manipulation over textual content is a core part of user workflows. Without text dragging functionality, users must resort to inefficient character-by-character or word-by-word selection through keyboard shortcuts, resulting in low productivity. Beyond efficiency considerations, text dragging ensures content consistency during

Figure 1: Illustration of the SCREENDRAG benchmark and the task of text dragging. The left part shows three levels of interface context within the benchmark (examples in Appendix G) and the right part shows the process of grounding the text selection by dragging.

cross-application transfers by preserving crucial metadata including font styles, formatting specifications, and spreadsheet formulas that would otherwise be lost when only raw text is reproduced. Moreover, many advanced features in productivity software, such as highlighting, annotating, and commenting, are only accessible after a span of text has been explicitly selected by dragging (an Adobe Acrobat example is shown in Figure 10). These considerations together underscore the importance of text dragging for GUI agents and motivate us to study it in this work.

Nevertheless, developing high-quality datasets for text dragging introduces unique methodological and data curation challenges. Existing data collection approaches are primarily designed for click-based grounding and rely heavily on HTML markup, which only provides coarse coordinates for interactive elements. In contrast, text dragging demands precise character-level coordinates that typically exceed the granularity available through standard HTML structures. While Xie et al. (2025) proposed to obtain fine-grained coordinates through application-specific scripts, this approach requires manual creation of source files, limiting its scalability. Furthermore, most screenshots from existing GUI grounding datasets exhibit insufficient textual density, rendering them inadequate for constructing meaningful and comprehensive training examples in text-rich scenarios.

To address this gap, we made the following contributions:

• We first carefully analyze and filter screenshots rich in textual content from existing datasets, and additionally collect 20K public academic paper-style documents. We further design an automated three-stage pipeline to synthesize text dragging examples directly from screenshots. In total, we curate 161K diverse and high-quality text dragging instances, which we term GUI-DRAG.

• To support systematic evaluation of text dragging capability, we further construct SCREENDRAG, a benchmark with 5,333 examples spanning different levels of interface context, along with three novel metrics to enable rigorous evaluation of text dragging capability.

• Our results on SCREENDRAG reveal a pronounced bias in existing grounding models toward click actions, including frontier systems such as the OpenAI Computer Use Agent. It underscores the need for the development of balanced datasets and grounding models that can reliably execute a broader set of actions, not limited to clicking.

• Our models, trained on Jedi-3B/7B through an efficient continual training strategy, achieve substantial improvements over the strongest baselines, demonstrating up to 18% absolute improvement (90% relative improvement) in accurately selecting the exact spans. Importantly, our models also preserve the base models' original click-based performance on benchmarks such as ScreenSpot, ScreenSpot-v2, and OSWorld-G by using only 10% original Jedi data.

## 2 METHODOLOGY

### 2.1 PROBLEM FORMULATION

GUI grounding refers to the process of mapping the natural language instruction into the mouse or keyboard actions. More formally, given the screenshot and the instruction, grounding model output

the action $a$, such as "click", "drag" or "type" with the corresponding parameters. Parameters can be coordinates (e.g., click$(x, y)$) or the text to be input (e.g., type(text)). Different from most works solely targeting the click action where the parameter is a single coordinate $(x, y)$, we instead focus on studying the text dragging capability where a pair of coordinates are required to ground the instruction by drag$(x_{\text{start}}, y_{\text{start}}, x_{\text{end}}, y_{\text{end}})$.

## 2.2 DATA CONSTRUCTION

Constructing datasets for text dragging presents several unique challenges that are not adequately addressed by existing GUI grounding efforts.

First, the development of prior datasets (Gou et al., 2025; Wu et al., 2024; Lu et al., 2024; Yu et al., 2025) heavily rely on the dual representation property of webpages, which provides correspondences between HTML and its rendered visual layout. This is effective for synthesizing click-based grounding data (e.g., clicking at a button), as such targets typically correspond to discrete, well-defined HTML elements. However, it is far less suitable for text dragging tasks, which require finer granularity. In particular, while dual representations contain bboxes for blocked units like paragraphs, they lack coordinates for finer spans such as individual sentences or multi-words, which do not exist as standalone elements. This limitation is critical, as selecting such fine-grained spans is a common and essential use case for text dragging. Recent work such as OSWorld-G (Xie et al., 2025) has attempted to overcome this limitation by application-specific scripts (e.g., in Word) to extract character-level coordinates. While it yields more precise supervision, it requires manually creating source files for each application, which is labor-intensive and hard to scale. To address these issues, we propose an automatic pipeline that synthesizes text dragging data directly from screenshots.

Beyond these methodological constraints, another challenge lies in the sources of training data. Many previous efforts (Xu et al., 2024; Yang et al., 2025) reuse screenshots from earlier datasets and apply their own processing pipelines for their purposes. However, existing GUI grounding datasets are primarily designed to support click-based interactions, with the goal of enabling agents to trigger icons, links, or navigational elements. As a result, they contain relatively little textual content and offer limited utility for constructing meaningful and diverse text-dragging examples. For instance, in the UGround (Gou et al., 2025) dataset, which crawls approximately 700K webpages, only about 0.7% of the collected screenshots contain more than 25 text-related tags (e.g., `<p>`, `<h1>`, `<h2>`). Although screenshots with sparse textual content may still be applicable for data synthesis, they fail to capture realistic usage scenarios where text dragging typically occurs in text-dense environments.

Table 1: Distribution of # text related tags in the Uground datasets.

| # of text tag | count (ratio) |
|---|---|
| > 200 | 7 (0.00%) |
| > 100 | 60 (0.01%) |
| > 50 | 594 (0.06%) |
| > 30 | 3991 (0.41%) |
| > 25 | 7991 (0.72%) |
| > 20 | 14625 (1.51%) |

To address this gap, we begin by filtering the UGround dataset to retain only screenshots containing at least 25 text-related tags, yielding approximately 8k images. Next, we manually review the Jedi training set and select screenshots that feature document-centric interfaces with substantial textual content, contributing an additional 2k examples. However, our preliminary exploration reveals that these two sources alone do not yield satisfactory performance in scenarios involving highly compact textual content. Therefore, we further collect ∼20k publicly available screenshots of paper-style documents[1] characterized by high text density and well-suited for text dragging scenarios.

To synthesize high-quality training data, we introduce a simple yet effective three-stage pipeline:

**Instruction Generation:** For each screenshot, we prompt o4-mini (OpenAI, 2025) (henceforth, the annotation model) to generate instructions and the corresponding target text spans referenced by the instruction (e.g., the instruction is 'Drag to select the last sentence' and the target text span is 'For drafting ... as Word.' in the Figure 2). To ensure broad coverage of different granularities of the text span, we include five granularities: **single sentence/paragraph**, **multiple sentences/paragraph**, and **multi-words span**. Here, a "sentence" is defined as a span terminated by appropriate punctuation (e.g., a period, exclamation mark, or question mark), rather than a line break.

---

[1] https://universe.roboflow.com/

To capture realistic usage patterns where people can use different ways to describe the same text span, we introduce five complementary categories of instructions: (1) **Semantic**: describe the span by meaning or topic (e.g., the paragraph introducing the Libre Office); (2) **Positional**: specify absolute or relative layout location (e.g., the second sentence of the first paragraph); (3) **Visual**: refer to visual appearance (e.g., the heading with bold text); (4) **Lexical**: anchor by literal content (e.g., sentence starting with the word 'For'); (5) **Compositional**: combine minimal cues from the four categories above. We intentionally prioritize positional and lexical categories for the instruction and granularities of sentence and multi-word level, as they are more aligned with how humans typically perform drag actions in real applications (e.g., commenting or highlighting in documents). A small-scale human study supports this design.

Each instruction is phrased in either an **explicit** form (e.g., "Drag to select...", "Drag the mouse to highlight...") where the drag action is directly specified or an **implicit** form (e.g., "Copy the range from...", "Highlight across..."), requiring the grounding model to infer that a drag action is needed.

**Grounding.** In this stage, we ground the instruction to pixel-level coordinates. We first apply OCR to the screenshot to obtain word-level bboxes. Given the OCR output and the target text span, the annotation model is used to identify the bboxes of the start and end words (i.e., $B_{\text{start}}$ and $B_{\text{end}}$) and retrieve the corresponding start and end coordinates $(x_{\text{start}}, y_{\text{start}}), (x_{\text{end}}, y_{\text{end}})$ accordingly. Empirically, we find that EasyOCR[2] performs reliably at the word level for our task. Additional details on the grounding process and EasyOCR hyperparameter choices are provided in Appendix E.

**Filtering.** To ensure data quality, we further go through two filtering processes. First, we annotate each screenshot with the selected start and end coordinates, i.e. the Set-of-Marks (SOM) technique (Yang et al., 2023). Then, we ask the annotation model to (1) verify that the instruction clearly corresponds to the intended span and (2) confirm that the annotation tightly enclose the target span. Second, we conduct manual spot checks on approximately 5% of the dataset to assess instruction clarity and annotation consistency. After filtering, we retain a final corpus of around 161k high-quality text dragging training examples, denoted as GUI-DRAG. The details on prompts used in three stages are provided in Appendix I.

## 2.3 TRAINING STRATEGY

The current dominant paradigm for training grounding models is collecting millions of examples and training a base model from scratch (Hong et al., 2024; Gou et al., 2025; Xu et al., 2024; Xie et al., 2025). While effective, this approach incurs substantial computational cost and time (e.g., Jedi-7B Xie et al. (2025) takes 1920 H100 hours). An alternative and more efficient strategy is continual training, where the model is further trained on a mixture of new data and part of its original data. It allows the model to acquire new skills while preserving its existing capabilities in an efficient manner. In our study, we adopt this efficient strategy to maintain the model's clicking capability while enhancing its text dragging performance. Specifically, we choose the Jedi-3/7B as our base model as it achieves competitive performance on various click-based grounding benchmarks while struggling to perform well on text dragging tasks. We randomly sample 10% of the original Jedi data, which leads to 750k Jedi training examples. Combined with our own 161k text dragging data, we train our own model, GUI-DRAG-3/7B, for two epochs. Notably, training GUI-DRAG-7B only takes around 350 H100 hours, achieving a substantial reduction in computational cost compared to training from scratch. We will conduct a more detailed analysis of how different amount of Jedi training data affects the model performance in Section 5. To assess the quality and utility of our text dragging dataset, we also train models exclusively on our own collected data, referred as Jedi-3B/7B (Drag). Additional implementation details are provided in Appendix D.

## 3 SCREENDRAG

In this section, we introduce SCREENDRAG along with three complementary metrics, which together provide an effective and rigorous evaluation framework. We use $\bar{X}$ and $\hat{X}$ to represent the prediction and ground truth, respectively, where $X$ can be coordinate or bbox.

---

[2]https://github.com/JaidedAI/EasyOCR

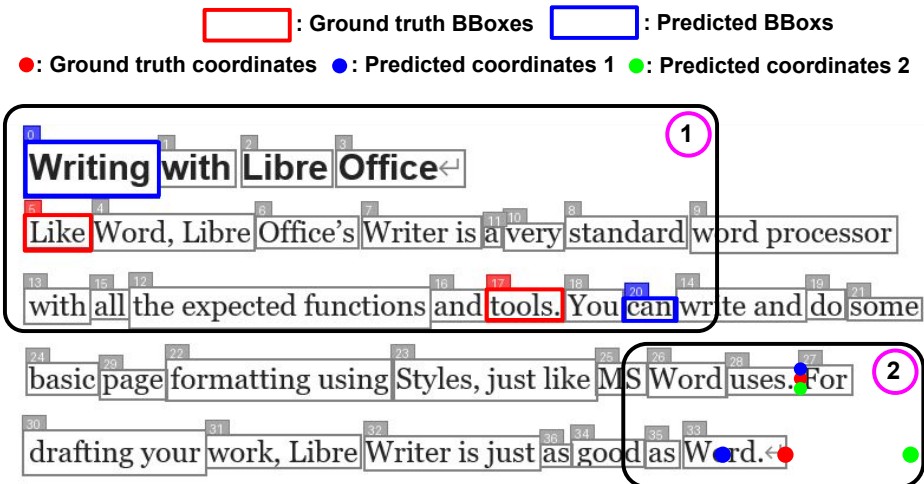

Figure 2: A screenshot with SOM in gray. In the top-left black box, the target text span is the first sentence, *"Like ... tools."*. Given the ground truth and predicted bboxes (The predicted start and end coordinates fall within bbox 0 and bbox 20, which we omit here to avoid clutter.), the B-Dist is 3 according to Equation 1. In the bottom-right box, the target span is the last sentence, *"For ... Word."*. Here, both predictions (blue and green) yield zero B-Dist, but only the green coordinates correctly capture the target span. The blue prediction fails because its $d_{\text{pixel}}$ does not satisfy the small threshold, whereas the green one succeed given the text snapping mechanism.

## 3.1 EVALUATION DATASET

To account for the fact that text dragging is a core part of workflows in productivity applications, we design our benchmark, SCREENDRAG, over Word, PowerPoint, and PDF readers. Concretely, we first manually collect and curate 110 multi-page documents across these three applications.

Nevertheless, GUI agents, particularly when equipped with the Model Context Protocol (MCP), can be deployed across a wide spectrum of usage scenarios by interfacing either directly with specific applications or with the operating system as a whole. Therefore, the screenshots that the grounding module receives can differ significantly in both scope and content as well. In certain deployment settings, the agent may only be provided with a tightly scoped document view (e.g., a single rendered page within a PDF reader) to support more targeted and efficient processing. In other cases, the agent is asked to perceive broader contexts by receiving the screenshot of the application window or the full desktop. To account for this diversity in agent implementation, we design our benchmark to include three levels of interface context: the document view, the application window, and the full desktop, as illustrated in Figure 1. This setup allows us to simulate a broad range of real-world use conditions and systematically evaluate grounding model performance under varying levels of contextual complexity.

For each manually captured screenshot, we first annotate individual words based on the OCR results. Human annotators are then instructed to randomly select text spans at different granularities by using different categories of instructions and label the corresponding ground truth start and end bounding boxes $(\hat{B}_{\text{start}}, \hat{B}_{\text{end}})$ using the screenshot with SOM. From these annotations, we derive the drag coordinates, $(\hat{x}_{\text{start}}, \hat{y}_{\text{start}})$ and $(\hat{x}_{\text{end}}, \hat{y}_{\text{end}})$, which are subsequently used to re-annotate the screenshot for another round of filtering process. In addition, we further use LLMs to classify each example as either **text-sparse** or **text-dense** (example in Appendix F). Text-sparse cases refer to the target span that is relatively isolated and therefore easier to select. By contrast, text-dense cases indicate that the target span is tightly surrounded by other selectable text, making precise dragging more challenging and placing higher demands on pixel-level accuracy. We also use LLMs to rephrase 20% of the instructions from explicit to implicit form to further diversify the benchmark. Detailed statistics including the resolution distribution for the benchmark are provided in Appendix B.

## 3.2 EVALUATION METRICS

**Drag Trigger Rate (DTR)**: This metric measures the proportion of cases where the model can successfully outputs correct drag action given the instructions. Note that models with different action

space have different definition for the drag action. For example, OpenAI CUA has a complete drag action: $\mathsf{drag}(x_{\text{start}}, y_{\text{start}}, x_{\text{end}}, y_{\text{end}})$, while Jedi and Qwen models need at least two actions which first output $\mathsf{click}(x_{\text{start}}, y_{\text{start}})$ or $\mathsf{move\_to}(x_{\text{start}}, y_{\text{start}})$ before executing $\mathsf{drag\_to}(x_{\text{end}}, y_{\text{end}})$. Our metric will account for such differences during evaluation.

**Bbox Dist (B-Dist)**: This is a metric defined at the bbox level. Specifically, we first map the predicted start and end coordinates to bboxes obtained from OCR using two criteria: (1) if a predicted coordinate lies inside a bbox, we directly assign it to that box; (2) if a predicted coordinate does not fall within any bbox, we assign it to the box with the closest horizontal distance. Based on the mapped $(\bar{B}_{\text{start}}, \bar{B}_{\text{end}})$, we compute the bbox-level distance as:

$$\text{B-Dist} = \tfrac{1}{2}\Big(\big|(\bar{B}_{\text{start}}) - (\hat{B}_{\text{start}})\big| + \big|(\bar{B}_{\text{end}}) - (\hat{B}_{\text{end}})\big|\Big). \tag{1}$$

where $|\cdot|$ denotes the index difference at the bbox level. During implementation, we find that OCR systems often parse bboxes in an arbitrary order (for example, bbox 10 and bbox 11 in Figure 2). To address this issue, we design a simple algorithm that automatically reorder the bboxes based on their spatial relationship, ensuring the effectiveness of calculating index differences.

**Success Rate (SR)**: While B-Dist captures misalignment at the bbox level, it remains a coarse measure and does not guarantee that the predicted span precisely selects the ground truth span (e.g. the predicted points in blue at Figure 2). To impose a stricter requirement, we introduce the SR metric, which evaluates whether the predicted span *exactly* matches the ground truth. In particular, when B-Dist $= 0$, we further assess the Euclidean distance between the predicted coordinates and the ground truth coordinates by computing:

$$d_{\text{pixel}} = \max\Big\{\big\|(\bar{x}_{\text{start}}, \bar{y}_{\text{start}}) - (\hat{x}_{\text{start}}, \hat{y}_{\text{start}})\big\|_2, \ \big\|(\bar{x}_{\text{end}}, \bar{y}_{\text{end}}) - (\hat{x}_{\text{end}}, \hat{y}_{\text{end}})\big\|_2\Big\}. \tag{2}$$

To ensure that the predicted coordinates are sufficiently close to the ground truth endpoints at the pixel level, it should satisfy $d_{\text{pixel}} < \phi$ where $\phi$ is a manually defined threshold.

However, an exception arises when either $\hat{B}_{\text{start}}$ or $\hat{B}_{\text{end}}$ lies at the beginning or end of a line. In these cases, predicted coordinates may fall slightly outside the ground truth span along the dragging direction but still yield a correct selection. This behavior is attributed to a common design feature in modern operating systems, known as *text snapping* (Miura & Saisho, 2014; Apple Inc., 2025; Microsoft, 2025), where selections will automatically extend to the nearest valid boundary once the pointer overshoots (e.g. the green prediction in Figure 2 can correctly select the target span given text snapping mechanism). To account for this effect, we integrate the snapping behavior into our SR metric to ensure the validity. Therefore, the SR can be formalized as:

$$\text{SR} = \begin{cases} 1, & \text{if B-Dist} = 0 \ \wedge \ \big(d_{\text{pixel}} < \phi \ \vee \ \text{text snapping}\big), \\ 0, & \text{otherwise.} \end{cases} \tag{3}$$

Taken together, these three novel metrics offer a set of comprehensive approaches to effectively and reliably measure the grounding model's text dragging capability.

## 4 EXPERIMENTS

### 4.1 SETUP

**Baselines:** We evaluate a range of frontier closed-source and open-source models that are widely used in GUI-related tasks. For closed-source models, we include OpenAI Computer Use Agent (CUA) (OpenAI., 2025) and Claude CUA (Anthropic., 2024). For each, we consider two configurations: the default setting, and a variant in which the system prompt explicitly indicates that drag actions are required (i.e., w/ hint). For open-source models, we focus on Qwen2.5-VL-3B/7B/32B (Bai et al., 2025), Jedi-3B/7B (Xie et al., 2025), and UI-TARS-1.5-7B (Qin et al., 2025). Although many other open-source grounding models exist, we restrict our evaluation to these for two reasons. First, most grounding models are trained exclusively for click-based interactions and thus cannot be meaningfully assessed on text dragging tasks. Second, while some models claim to support a broader action space beyond clicking, our preliminary experiments show that they consistently fail to produce valid drag actions, yielding a DTR of zero. Consequently, we omit them from further evaluation.

Table 2: Performance on SCREENDRAG with breakdown on text-sparse, text-dense settings. The best and second-best results under each metric are **bolded** and underlined. * indicates that the model has a standalone complete drag action.

| Experimental Setting | DTR↑ | Text-Sparse | | Text-Dense | | Avg. (Total) | |
|---|---|---|---|---|---|---|---|
| | | B-Dist↓ | SR↑ | B-Dist↓ | SR↑ | B-Dist↓ | SR↑ |
| **Open-Source** | | | | | | | |
| Qwen2.5-VL-3B | 5.0% | 44.5 | 3.2% | 43.8 | 0.0% | 44.3 | 2.26% |
| Qwen2.5-VL-7B | 5.0% | 27.7 | 3.35% | 30.8 | 1.16% | 28.7 | 2.64% |
| Qwen2.5-VL-32B | 55.8% | 23.5 | 5.84% | 27.1 | 1.65% | 24.4 | 5.09% |
| Jedi-3B | 94.1% | 12.1 | 19.0% | 17.2 | 8.53% | 13.4 | 16.3% |
| Jedi-7B | 77.5% | 14.3 | 12.1% | 18.3 | 4.99% | 15.4 | 10.3% |
| UI-TARS-1.5-7B* | 84.6% | 13.0 | 23.6% | 19.5 | 9.36% | 14.7 | 20.0% |
| **Closed-Source** | | | | | | | |
| OpenAI CUA* | 85.7% | 9.70 | 21.4% | 12.98 | 8.13% | 10.1 | 16.0% |
| Claude CUA* | 47.4% | 10.44 | 17.0% | 8.92 | 12.59% | 10.5 | 18.1% |
| OpenAI CUA (w/ hint)* | 91.7% | 8.68 | 18.0% | 12.83 | 6.74% | 9.15 | 16.1% |
| Claude CUA (w/ hint)* | 96.9% | 8.63 | 16.9% | 10.74 | 11.39% | 9.73 | 16.6% |
| **Ours** | | | | | | | |
| Jedi-3B (Drag) | **100.0**% | 7.9 | 39.7% | 9.2 | 20.1% | 8.2 | 34.7% |
| Jedi-7B (Drag) | **100.0**% | 7.4 | 36.1% | 8.8 | 16.6% | 7.7 | 31.2% |
| GUI-DRAG-3B | **100.0**% | 6.9 | **43.6**% | 7.2 | **22.9**% | 7.0 | **38.1**% |
| GUI-DRAG-7B | **100.0**% | **6.2** | 38.1% | **6.7** | 19.8% | **6.4** | 33.1% |

**Evaluation:** We evaluate models using the three metrics introduced in Section 3, each designed to capture different aspects of text dragging performance. For the B-Dist and SR metrics, we only consider cases where the model can accurately output the drag action. For the SR metric, we set the threshold $\phi$ to 3 pixels. This value is empirically determined by manually inspecting 100 examples, and is found to strike a good balance between reducing false positives and false negatives.

## 4.2 RESULTS

**Main Findings:** Across both text-sparse and text-dense settings, our models consistently outperform all baselines. In particular, GUI-DRAG-3B achieves 43.6% SR on text-sparse and 22.9% SR on text-dense inputs, representing absolute improvements of 20% and 10% over the best-performing baselines, respectively. Moreover, our models obtain substantially lower B-Dist on average, indicating closer alignment with ground truth spans and thus better control and understanding of text-drag operations. Among open-source baselines, UI-TARS-1.5-7B achieves the highest SR and even surpasses the closed-source models; however, its relatively high B-Dist suggests systematic failures in specific scenarios despite overall strong performance. Upon closer examination, we find that it frequently fails in cases where instructions refer to the text spans at sentence granularity by using positional cues, particularly under text-dense conditions. This may indicate limited coverage in their training data. Meanwhile, closed-source models generally outperform other open-source baselines but the performance drop substantially under the more challenging text-dense setting and largely lag behind our models trained with GUI-DRAG.

Besides, we find that incorporating the original Jedi data, despite being click-dominant, further improves drag performance. We hypothesize that this is due to the shared requirement in both tasks to ground instructions into fine-grained pixel-level coordinates. Hence, the implicit grounding signals in click data can benefit drag localization as well.

Additional results with detailed breakdowns across interface contexts, task categories, and span granularities are provided in Appendix H. Notably, we observe that grounding models exhibit distinct performance trends across interface levels, suggesting that GUI agents may need to adaptively select

grounding models depending on the usage scenario and the specific MCP configuration. Taken together, the findings over SCREENDRAG highlight the limitations of existing grounding models in handling text dragging and validate the effectiveness of our data collection pipelines.

**Biasing Towards Clicking:** Surprisingly, we find that baseline models often fail to reliably trigger the drag action, even when dedicatedly trained for computer use scenarios. For instance, OpenAI CUA and UI-TARS-1.5-7B only reach a DTR of 85.7% and 84.6%, respectively. To dive deeper, we further analyze their output action distributions across implicit and explicit instructions in Figure 3. The result shows that models exhibit a pronounced bias toward click actions; when the instruction directly requires a drag operation, models still frequently persist with the click action. Even with an additional hint in the system prompt, OpenAI CUA, known for strong instruction following, still fails to elicit the correct drag action perfectly. Such issues become more

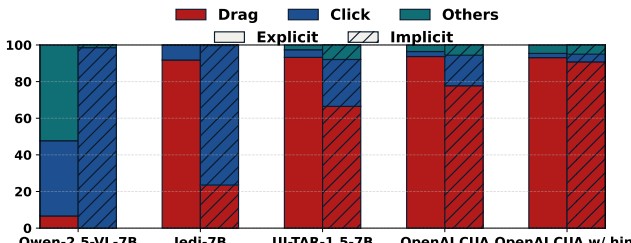

Figure 3: Distribution of actions across explicit and implicit instructions across 5 models.

severe under the implicit instruction setting, where the models like Qwen-2.5-VL-7B and Jedi-7B intend to blindly output click actions. This phenomenon raises the questions about whether current grounding models possess sufficient instruction understanding and underscores a critical limitation that all models are severely biased towards click actions. It underscores the importance of balancing training datasets and advancing generalist grounding models that can robustly interpret and perform a wider spectrum of actions beyond clicking.

## 5 ANALYSIS

**Continual Training**: We evaluate the impact of incorporating varying proportions of Jedi data through a continual training approach on both click and text dragging tasks. For assessing the click performance, we employ three widely adopted click-based benchmarks (ScreenSpot-Pro, ScreenSpot-V2, and OSWorld-G) and report the averaged scores across these three in Figure 4 (detailed breakdowns provided in Appendix C). Overall, our results suggest that the scaling law does not straightforwardly hold under continual training, and the training dynamics vary across model sizes. For the 7B model, incorporating higher proportions of Jedi data effectively preserves click performance, while the 3B model maintains robust performance with minor degradation when utilizing 15% original data. Moreover, more Jedi data will hinder the acquisition of text dragging capability for the 7B model, which is not observed in the 3B model. To optimize the tradeoff between click and drag task performance while maintaining computational efficiency, we thus integrate 10% of the Jedi data for training our GUI-DRAG-3/7B. While these findings demonstrate the efficiency and the promise of continual training for developing more generalist grounding models, future work should further investigate how to perform continual training in a more reliable and systematic manner.

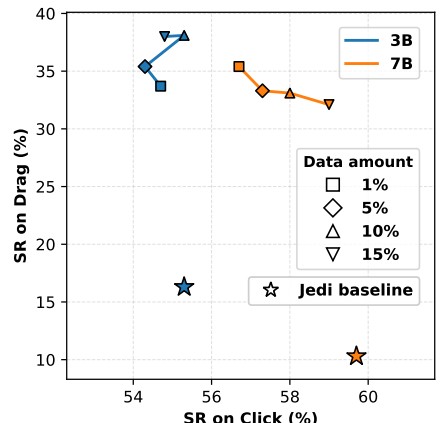

Figure 4: Success rate on click-based benchmarks (x-axis; average score reported) and SCREENDRAG (y-axis) with different proportion of Jedi data.

**Case Study on OSWorld**: To further evaluate whether our model can benefit more challenging agentic tasks, we carefully analyze the OSWorld benchmark (Xie et al., 2024) and identify three tasks where two human annotators agree that text dragging is both necessary and more efficient than alternative approaches to complete the tasks. Details on these tasks are provided in Appendix K. For GUI-DRAG-7B, we employ o3 (OpenAI, 2024) as the high-level planner to generate instructions, with GUI-DRAG-7B serving to translate these instructions into low-level actions.

Our results in Table 3 demonstrate that GUI-DRAG-7B successfully completes all three tasks, showcasing its effective text dragging capabilities. While Claude CUA and OpenAI CUA complete 2 out of 3 tasks, closer examination reveals that they frequently resort to unconventional shortcuts such as triple-clicking and quadruple-clicking to select sentences and paragraphs. Although these idiosyncratic techniques prove effective in specific contexts, they rely on domain specific knowledge that may not generalize to other applications and are less intuitive compared to direct drag actions. In contrast, our model's ability to perform natural drag operations represents a more robust and transferable approach to text selection across diverse GUI environments.

Table 3: Success rate on three examples from OSWorld.

| Model | OpenAI CUA | Claude CUA | o3 | o3 + GUI-DRAG-7B |
|---|---|---|---|---|
| **SR** | 2/3 | 2/3 | 0/3 | 3/3 |

## 6 RELATED WORK

**GUI Grounding.** GUI grounding, the task of mapping natural language instructions to executable GUI operations, is a fundamental capability for LLM-based GUI agents (Nguyen et al., 2024; Liu et al., 2025a). Early approaches primarily rely on textual representations such as HTML or accessibility trees, which allow models to select from predefined textual elements (Deng et al., 2023) or bounding boxes drawn on images (Yang et al., 2023; Zheng et al., 2024; He et al., 2024; Yan et al., 2023), which, despite effective in constrained setting, suffer from incompleteness, noise, and computational overhead. To address these issues, recent work has shifted toward vision-only methods, where MLLMs predict actions directly from screenshots (Cheng et al., 2024; Hong et al., 2024; Gou et al., 2025; Lin et al., 2024; Yu et al., 2025; Yang et al., 2024; Lu et al., 2024), which is now widely adopted in recent GUI agents (Qin et al., 2025; Wang et al., 2025; OpenAI., 2025; Anthropic., 2024) and is even incorporated as a core ability of general-purpose MLLMs (Wang et al., 2024; Bai et al., 2025). Despite the progress, existing efforts mostly focus on click-based grounding, leaving dragging as a largely unsolved challenge. Existing models and agents, either exclude it from their action spaces, or actually fail to perform it reliably, as highlighted by our experiments.

**Datasets and Benchmarks.** To facilitate progress in GUI grounding research, numerous datasets (Lin et al., 2024; Gou et al., 2025; Yang et al., 2024; Xu et al., 2024; Zhang et al., 2025) and benchmarks (Cheng et al., 2024; Nayak et al., 2025; Li et al., 2025) have been developed. Among benchmarks, ScreenSpot (Cheng et al., 2024) was the first to isolate GUI visual grounding as a standalone task. ScreenSpot-Pro (Li et al., 2025) and OSWorld-G (Xie et al., 2025) further expand the evaluation to more challenging tasks such as professional application use. However, dragging remains largely underexplored. While recent work, UI-Vision (Nayak et al., 2025), accesses the performance of moving objects by dragging, text dragging scenarios are not studied. With respect to datasets, only a few works targeting general-purpose GUI agents include drag-related data (Qin et al., 2025; Wang et al., 2025) and the coverage is relatively limited and not open-sourced. To address these gaps, our benchmark SCREENDRAG and dataset GUI-DRAG are specifically designed to advance research on text dragging in GUI environments.

## 7 CONCLUSION

In this work, we first introduce a scalable pipeline to automatically synthesize text dragging examples directly from screenshots. Building upon this pipeline, we construct GUI-DRAG, the first dataset specifically designed to enhance text dragging performance in GUI grounding models. Additionally, we develop the SCREENDRAG benchmark alongside three novel evaluation metrics that collectively enable systematic and rigorous evaluation of text dragging capability. Using an efficient continual training strategy, our model achieves substantial improvements over existing grounding models while preserving its original click performance. By releasing the complete recipe including datasets, benchmark and the models, we hope our study serves as a starting point and motivates future research to investigate broader and generalist GUI grounding beyond just clicking.

ETHICS STATEMENT

Our work synthesizes GUI-DRAG using publicly available screenshots from prior datasets, including Jedi (Apache-2.0 license), UGround (CC-BY-NC-SA 4.0 license), and a paper-document dataset from Universe[3] (MIT license). All data are used in strict accordance with their respective licenses. Since these datasets are publicly released for research purposes, they do not raise additional ethical or legal concerns. The trained model is developed solely to enhance text dragging within GUI grounding and is intended for beneficial applications, thereby posing no further ethical risks. In addition, we construct SCREENDRAG by manually collecting screenshots from public websites, ensuring that the data are license-compliant and free from usage restrictions.

REPRODUCIBILITY STATEMENT

We provide details on the composition of the training data, the training pipeline and process, as well as the construction of the benchmark and the definition of evaluation metrics in Section 2 and Section 3. We also describe the design choices behind the baseline models and implementation details of the results in Section 4 and Section 5. All related materials, including code and data, will be open-sourced upon acceptance.

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

Beyond Click: A Step Towards Generalist Grounding Models

Table of Contents for Appendix.

## A    LLM Usage Statement

LLMs are mainly used in places below. First, our data collection pipeline is powered by LLMs. Second, human annotators are allowed to use LLMs to speed up the annotation process. Third, we use LLMs to classify the examples into text-sparse and text-dense. Fourth, we use LLMs to rephrase 20% of the instructions from explicit format to implicit format to further diversify the benchmark. Fifth, we use LLMs to refine word choices and polish the writing.

## B    Benchmark Statistics

We include the statistics of the benchmark in Table B and the overall screenshot resolution in Figure 5. Among the 5,333 examples, 3,998 belong to the text-sparse subset and 1,335 belong to the text-dense subset.

Table 4: Detailed statistics for three types of file in the benchmark.

| Metric | pdf | pptx | docx |
|---|---|---|---|
| *Format* | | | |
| explicit | 1436 | 1296 | 1172 |
| implicit | 489 | 467 | 473 |
| *Granularity* | | | |
| sentence | 901 | 731 | 1011 |
| multi-words | 455 | 860 | 331 |
| multi-sentence | 320 | 145 | 130 |
| paragraph | 237 | 22 | 172 |
| multi-paragraph | 12 | 5 | 1 |
| *Category* | | | |
| positional | 752 | 684 | 658 |
| semantic | 347 | 301 | 313 |
| lexical | 293 | 289 | 283 |
| visual | 267 | 246 | 144 |
| compositional | 266 | 243 | 247 |

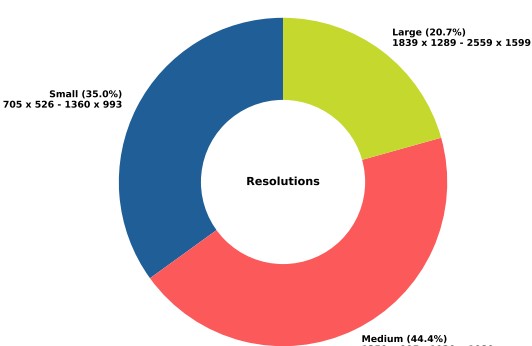

Figure 5: Resolution size analysis of the benchmark.

## C  CLICK-BASED BENCHMARK RESULTS

To ensure reliable reproduction, we rerun the results for Jedi models independently using the officially released checkpoints. The results, summarized in Table 5, report the Success Rate (SR), defined as the proportion of predictions whose predicted coordinate lies within the ground truth bbox.

Table 5: Performance on OSWorld-G, ScreenSpot Pro, ScreenSpot-v2.

| Model | OSWorld-G | ScreenSpot-Pro | ScreenSpot-v2 |
|---|---|---|---|
| Jedi-3B | 0.47 | 0.32 | 0.87 |
| Jedi-3B (GUI-DRAG + 1% Jedi data) | 0.45 | 0.32 | 0.87 |
| Jedi-3B (GUI-DRAG + 5% Jedi data) | 0.45 | 0.31 | 0.87 |
| Jedi-3B (GUI-DRAG + 10% Jedi data) | 0.46 | 0.32 | 0.88 |
| Jedi-3B (GUI-DRAG + 15% Jedi data) | 0.45 | 0.32 | 0.87 |
| Jedi-7B | 0.55 | 0.33 | 0.91 |
| Jedi-7B (GUI-DRAG + 1% Jedi data) | 0.51 | 0.29 | 0.90 |
| Jedi-7B (GUI-DRAG + 5% Jedi data) | 0.51 | 0.31 | 0.90 |
| Jedi-7B (GUI-DRAG + 10% Jedi data) | 0.53 | 0.32 | 0.89 |
| Jedi-7B (GUI-DRAG + 15% Jedi data) | 0.54 | 0.32 | 0.90 |

## D  TRAINING DETAILS

We put the detailed training hyperparameters in Table 6.

Table 6: Training hyperparameters for training.

| Hyperparameter | Value |
|---|---|
| max_pixels | 2116800 |
| min_pixels | 3116 |
| per_device_train_batch_size | 4 |
| gradient_accumulation_steps | 2 |
| learning_rate | 1.0e-5 |
| num_train_epochs | 2.0 |
| lr_scheduler_type | cosine |
| warmup_ratio | 0.1 |
| bf16 | True |

## E  DETAILS ON THE GROUNDING PROCESS

After getting the $\hat{B}_{\text{start}}$ and $\hat{B}_{\text{end}}$, we can subsequently conduct ground truth drag coordinates $(\hat{x}_{\text{start}}, \hat{y}_{\text{start}}, \hat{x}_{\text{end}}, \hat{y}_{\text{end}})$. Specifically, we use the middle point of the left edge of $\hat{B}_{\text{start}}$ and the middle point of the right edge of $\hat{B}_{\text{end}}$ as the start and end coordinates.

For the EasyOCR hyperparameters used, we disable paragraph grouping to keep word-level boxes and tune thresholds as listed in Table 7 to improve word localization for drag spans.

Table 7: EasyOCR hyperparameters.

| Hyperparameter | Value |
| --- | --- |
| paragraph | False |
| text_threshold | 0.7 |
| width_ths | 0.1 |
| bbox_min_size | 1 |
| min_size | 5 |

## F  TEXT-SPARSE AND TEXT-DENSE EXAMPLES

We put the text-dense and text-sparse examples in Figure 6. Both target text spans are at the sentence granularity.

A novel may have any number of climaxes, each perhaps a little more intense than the one

preceding, so that the effect is of being swept over ever-higher foothills to the highest peaks at

the end.↵

↵
The setting of a novel is likely to jump from one place to another.↵
↵
There may be sudden shifts from one period of time to another. The action may jump back to an earlier date (flashbacks – which may explain later actions that otherwise could not be understood). When several plots are being pursued, the author may ask us to follow one plot through a certain period of time, then return to pick up a second plot and carry it through the same time period.↵
↵
In a novel, the point of view from which the story is told is likely to shift from person to person.↵
↵
The length of a novel makes it likely that several themes will be illustrated in the course of the action. ↵
↵
↵

Figure 6: Target text span that belongs to text-sparse category is marked in blue, while the one that belongs to text-dense category is marked in red.

## G  INTERFACE CONTEXT EXAMPLES

We put three examples of the same file in different interface context in Figure 7, Figure 8, and Figure 9. Each corresponds to the document view, application window, and desktop view, respectively.

The history of photo ethics in journalism has been studied earlier in Western academia. As early as 1964, knowledge was mentioned in Western journalism textbooks, and Curtis Mac Dougall's book *The Press and Its Problems* stated that photographers should give more consideration to the ethical issues they face in their work (MacDougall, 1964). In 1978, Horold Evans identified four areas of ethical concern: violence, invasion of privacy, sexuality and public morality, and photo falsification(Harold，1978).↵

↵

With the advent of the "picture age," the role of pictures in news reporting has become more prominent, and the live and visual nature of pictures has led to an unprecedented development of news pictures. However, the ensuing problems have also aroused people's concern, such as some media or journalists creating false pictures for their own purposes, the *South China tiger fake* photos have aroused

Figure 7: Document view example.

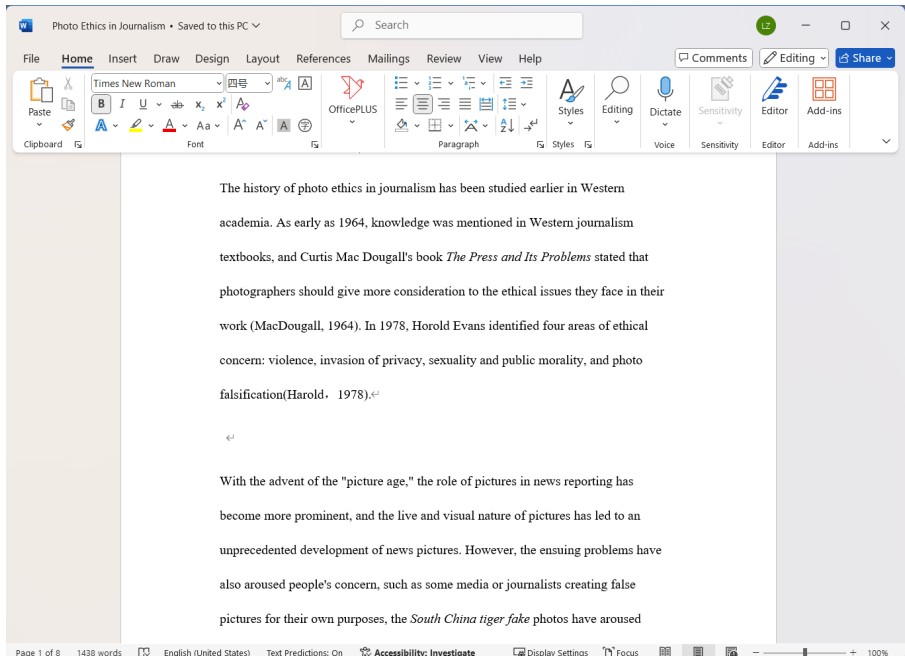

Figure 8: Application window example.

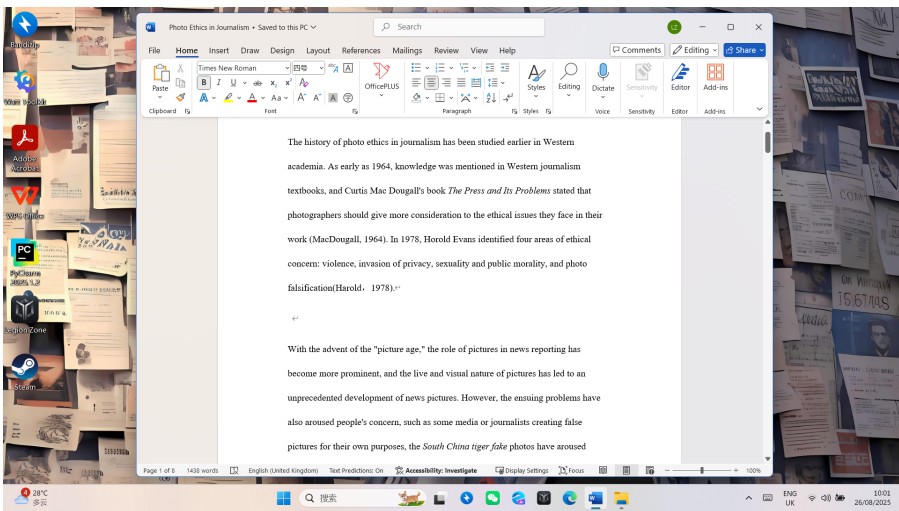

Figure 9: Desktop view example.

# H GRANULARITY BREAKDOWN

We put the performance with breakdowns in different granularities, categories and different levels of interface context in Appendix H.1, Appendix H.2, and Appendix H.3.

## H.1 GRANULARITY BREAKDOWN

Performance breakdown across different granularities in Table 8.

Table 8: Performance on SCREENDRAG with breakdown on different granularities.

| Model | Multi-paragraph | | Multi-sentence | | Multi-words | | Paragraph | | Sentence | |
|---|---|---|---|---|---|---|---|---|---|---|
| | B-Dist↓ | SR↑ | B-Dist↓ | SR↑ | B-Dist↓ | SR↑ | B-Dist↓ | SR↑ | B-Dist↓ | SR↑ |
| Open-Source | | | | | | | | | | |
| Qwen2.5-VL-3B | 58.00 | 0.00% | 81.50 | 0.00% | 25.16 | 7.14% | 50.16 | 0.00% | 44.01 | 0.76% |
| Qwen2.5-VL-7B | 66.67 | 0.00% | 31.43 | 0.00% | 27.26 | 6.82% | 38.64 | 0.00% | 25.88 | 2.63% |
| Qwen2.5-VL-32B | 25.75 | 8.33% | 41.14 | 3.92% | 17.06 | 5.94% | 34.93 | 8.66% | 23.16 | 4.23% |
| Jedi-3B | 27.97 | 16.67% | 18.91 | 19.55% | 7.54 | 17.83% | 24.40 | 22.99% | 13.71 | 13.65% |
| Jedi-7B | 35.44 | 23.53% | 21.50 | 16.63% | 8.80 | 8.64% | 27.94 | 16.86% | 15.80 | 8.64% |
| UI-TARS-1.5-7B | 13.06 | 17.65% | 16.63 | 27.40% | 8.00 | 20.78% | 18.89 | 35.92% | 17.80 | 15.29% |
| Closed-Source | | | | | | | | | | |
| OpenAI CUA | 26.97 | 23.53% | 12.21 | 24.90% | 7.61 | 18.98% | 17.12 | 26.04% | 10.66 | 14.63% |
| Claude CUA | 19.55 | 0.00% | 18.30 | 21.74% | 7.45 | 11.15% | 20.91 | 29.19% | 8.53 | 17.28% |
| OpenAI CUA (w/ hint) | 18.85 | 23.53% | 13.48 | 21.61% | 6.21 | 17.38% | 15.04 | 26.05% | 10.31 | 13.41% |
| Claude CUA (w/ hint) | 31.64 | 5.56% | 14.14 | 21.83% | 6.91 | 11.67% | 17.87 | 31.95% | 7.91 | 15.09% |
| Ours | | | | | | | | | | |
| Jedi-3B (Drag) | 20.14 | 44.44% | 9.87 | 30.67% | 6.89 | 39.81% | 13.89 | 44.54% | 6.52 | 28.67% |
| Jedi-7B (Drag) | 12.89 | 33.33% | 10.56 | 32.27% | 6.09 | 36.21% | 14.32 | 38.52% | 6.98 | 26.64% |
| GUI-DRAG-3B | 21.64 | 33.33% | 9.83 | 43.53% | 6.87 | 44.23% | 9.64 | 50.81% | 5.86 | 31.18% |
| GUI-DRAG-7B | 12.03 | 44.44% | 9.24 | 38.15% | 6.08 | 37.30% | 8.11 | 50.35% | 5.56 | 26.33% |

## H.2 CATEGORY BREAKDOWN

Performance breakdown across different instruction categories in Table 9.

Table 9: Performance on SCREENDRAG with breakdown on different categories.

| Model | Compositional | | Lexical | | Positional | | Semantic | | Visual | |
|---|---|---|---|---|---|---|---|---|---|---|
| | B-Dist↓ | SR↑ | B-Dist↓ | SR↑ | B-Dist↓ | SR↑ | B-Dist↓ | SR↑ | B-Dist↓ | SR↑ |
| **Open-Source** | | | | | | | | | | |
| Qwen2.5-VL-3B | 28.57 | 9.09% | 43.62 | 0.00% | 49.45 | 2.01% | 42.71 | 2.63% | 12.20 | 0.00% |
| Qwen2.5-VL-7B | 10.72 | 0.00% | 69.58 | 3.85% | 27.42 | 2.50% | 16.23 | 0.00% | 33.39 | 10.53% |
| Qwen2.5-VL-32B | 21.35 | 5.32% | 23.59 | 3.74% | 25.46 | 5.23% | 25.47 | 4.65% | 23.32 | 6.41% |
| Jedi-3B | 10.53 | 14.88% | 7.20 | 14.69% | 17.76 | 16.07% | 13.62 | 17.80% | 10.52 | 18.95% |
| Jedi-7B | 15.33 | 9.14% | 9.13 | 5.96% | 20.35 | 9.95% | 13.91 | 15.45% | 9.46 | 10.48% |
| UI-TARS-1.5-7B | 10.85 | 18.43% | 10.86 | 13.84% | 19.42 | 18.23% | 13.98 | 25.81% | 9.55 | 27.10% |
| **Closed-Source** | | | | | | | | | | |
| OpenAI CUA | 8.22 | 16.64% | 8.78 | 15.35% | 12.70 | 16.36% | 8.94 | 22.55% | 10.45 | 22.39% |
| Claude CUA | 10.04 | 11.80% | 9.18 | 14.97% | 10.16 | 17.49% | 12.34 | 20.23% | 8.91 | 13.41% |
| OpenAI CUA (w/ hint) | 9.13 | 16.21% | 8.67 | 13.23% | 11.27 | 15.46% | 8.70 | 20.48% | 8.42 | 19.17% |
| Claude CUA (w/ hint) | 7.77 | 14.05% | 10.51 | 13.87% | 8.19 | 17.14% | 11.61 | 18.97% | 8.59 | 13.60% |
| **Ours** | | | | | | | | | | |
| Jedi-3B (Drag) | 6.32 | 31.85% | 6.18 | 30.12% | 8.75 | 32.90% | 6.08 | 36.88% | 9.45 | 40.25% |
| Jedi-7B (Drag) | 6.65 | 29.10% | 6.49 | 27.40% | 9.19 | 30.18% | 6.40 | 34.13% | 7.81 | 37.60% |
| GUI-DRAG-3B | 5.21 | 36.38% | 5.13 | 32.95% | 7.78 | 37.34% | 5.47 | 42.04% | 11.08 | 44.14% |
| GUI-DRAG-7B | 6.00 | 31.88% | 4.31 | 27.86% | 6.79 | 32.38% | 6.24 | 37.88% | 8.24 | 36.23% |

## H.3 INTERFACE CONTEXT BREAKDOWN

Performance breakdown across different levels of interface contexts in Table 10.

Table 10: Performance on SCREENDRAG with breakdown on different interfaces context.

| Model | Document | | APP | | Desktop | |
|---|---|---|---|---|---|---|
| | B-Dist↓ | SR↑ | B-Dist↓ | SR↑ | B-Dist↓ | SR↑ |
| **Open-Source** | | | | | | |
| Qwen2.5-VL-3B | 13.45 | 10.26% | 45.00 | 0.63% | 60.49 | 1.47% |
| Qwen2.5-VL-7B | 20.71 | 3.04% | 39.56 | 0.00% | 173.14 | 0.00% |
| Qwen2.5-VL-32B | 8.26 | 5.58% | 20.87 | 5.59% | 43.62 | 3.96% |
| Jedi-3B | 9.80 | 16.05% | 13.95 | 13.68% | 17.19 | 19.83% |
| Jedi-7B | 10.58 | 10.08% | 15.20 | 10.84% | 21.11 | 9.89% |
| UI-TARS-1.5-7B | 11.10 | 18.56% | 15.04 | 20.52% | 18.98 | 21.25% |
| **Closed-Source** | | | | | | |
| OpenAI CUA | 7.29 | 19.93% | 10.79 | 17.18% | 14.36 | 16.81% |
| Claude CUA | 6.02 | 15.84% | 10.91 | 17.88% | 15.18 | 14.61% |
| OpenAI CUA (w/ hint) | 6.85 | 17.48% | 9.85 | 16.73% | 13.43 | 15.19% |
| Claude CUA (w/ hint) | 6.65 | 15.36% | 8.92 | 17.72% | 12.43 | 15.07% |
| **Ours** | | | | | | |
| Jedi-3B (Drag) | 4.85 | 32.80% | 7.62 | 37.25% | 10.12 | 32.85% |
| Jedi-7B (Drag) | 5.15 | 30.13% | 7.88 | 35.00% | 10.61 | 28.16% |
| GUI-DRAG-3B | 4.07 | 36.56% | 7.75 | 39.55% | 9.57 | 38.55% |
| GUI-DRAG-7B | 3.16 | 35.67% | 7.22 | 34.28% | 9.20 | 28.47% |

# I PROMPTS

We include three types of prompts used in our pipeline in Appendix I.1, Appendix I.2, and Appendix I.3.

## I.1 PROMPTS FOR INSTRUCTION GENERATION

```
You are given a screenshot input. Your task is to generate natural
    language referring expressions that specify different target
    text spans contained within the screenshot where users
    typically perform mouse drag actions for selection. Focus
    exclusively on selectable text content and ignore non-text
    elements, non-selectable areas, or elements that users don't
    commonly select in daily usage (e.g., placeholders within input
    fields, clickable UI elements such as toolbar icons or buttons).

Below are the five categories of referring expressions with their
    corresponding definitions and examples.

\#\# Semantic

Definition: describe the target text span based on its meaning,
    intent, or topical content.

Drag to select the paragraph discussing how to download models.
Using drag to highlight the paragraphs that infer the causes of
    failure.
Highlight the sentence about Kobe Bryant's career by dragging.
Drag the mouse to select consecutive words referring to the weight
    of the MacBook Pro.
highlight across the list items showing the D.O.B. of the
    characters in the movie "The Lord of the Rings".

\#\# Positional

Definition: refer to selecting text or elements based on their
    spatial or structural location within the document. This
    includes absolute positioning (using ordinal numbers or
    directional indicators like "third paragraph "last sentence
    "top of page") and relative positioning (location relative to
    other elements like "text below Figure 1 "words left of the
    login button").

Drag to select the second last paragraph at the bottom of the page.
Highlight the last three lines by using drag in the code blocks.
Highlight the content of the sentence immediately below the chart
    title.
Select the exact text span showing the words on the left side of
    the login button.
Select and copy the third sentence of the first paragraph.
Select all rows from row 1 to row 10 (inclusive) in the spreadsheet
    (include the row headers).
Select first sentence in the top-right corner of the page by
    dragging.
Drag the second sentence of the 2nd paragraph.
Drag the last sentence of the last paragraph.
Drag to select the 4th and 5th sentences of the first paragraph.

\#\# Visual

Definition: refer to distinctive visual features of the text, such
    as style, font color, size, emphasis, or highlighting.
```

Drag to highlight the paragraph written in bold italics.
Select all the paragraphs highlighted in yellow.
Copy the sentence in red font.
dragging to select the words with the largest font size on the
    screen.
Select all the words within the grey block by dragging.

\#\# Lexical

Definition: refer to the text by referencing its literal or quoted
    content, including the starting words, key phrases, or exact
    match.

Select the range of the sentence ending with 'before submission is
    due'.
Drag to highlight the paragraph that begins with "To get started
    with Python...".
Highlight and copy the sentence containing the phrase "AI is
    transforming industries".
Highlight across the words that say 'Monday, Tuesday, and so on'.
Select the text span starting with "This photo" and ending with
    "happy" by dragging.
Select to copy the content starting with character 'c' and ending
    with character 'e'.

\#\# Compositional

Definition: refer to the composition of the four categories
    mentioned above. You can randomly select and combine the
    features of the four categories above to generate a referring
    expression.

Drag to highlight the paragraph written in bold italics, discussing
    the usage of the model.
Select to copy the paragraphs which are highlighted in yellow and
    positioned at the top of the page.
Copy the sentence in red font, starting with the word 'AI'.
Drag the mouse to select the second last blue text span.

**Task Requirements**

Generate referring expressions for each of the five categories
    (semantic, positional, visual, lexical, and compositional)
    sequentially. For each category, you must:

1. You should first reason about easibility of generating a
    suitable referring expression for that category. It is normal
    for some categories to have no suitable expressions for certain
    screenshots. For example, not all screenshot contain salient
    visual features. To ensure high-quality generation, you could
    just set the availability to false if generating expressions
    under such category is unsuitable.

2. If feasible, then you should continute yhe step 3 to help with
    generating the referring expression. If not, you can leave
    empty to the left fields and don't need to continue.

3. If the category is about visual feature, you have to identify
    the most salient features under this category from the
    screenshot. For other categories, you should try to focus on
    areas which are text-dense. For example, it would be great to
    have target text span locating in a paragraph, etc. After that,
    you should both generate a referring expression and the target
    text span indicated by the referring expression. For target

text span, never omit the details of the full text span even if
the span is very long. This is because the post-process will
need the full content fo the target text span.

*Requirements when generating the target text span*:

The extracted text must include all punctuation marks and special
characters exactly as they appear in the screenshot. Even if
the text span in the screenshot contain certain style or font,
you only need to generate the pure text.

Extract the complete text span including all punctuation marks
(periods, commas, quotation marks, etc.) exactly as shown. Also
follow the left-to-right then top-to-bottom order, which is
exactly the same order for human reading. Always remember to
add the correct puncuation marks at the end if the target text
span is about sentence(s) or paragraphs.

Essentially, this is asking you to do the OCR correctly.

The target text span can be these granularities:
- Single or multiple paragraphs
- Single or multiple sentences
- Multiple consecutive words (single words typically don't require
dragging)

Note that the sentence should be ended with a punctuation mark like
period, exclaimation mark or question mark. Comma should not be
treated as the end of the sentence.

*Requirements when generating the referring expression*:

Generate expressions that are clear and specific enough while not
too wordy, that only the target text span you extracted can
match.

When generating compositional referring expressions, combine only
the minimum necessary features from different categories to
uniquely identify the target text span.

Use either the explicit or implicit approach to generate the
referring expression. More specifically:

\#\# Expressing Dragging Actions: Explicit vs. Implicit Approaches

Ensure users understand that a mouse drag action is required by
using both explicit and implicit approaches across different
expressions:

**Explicit Approaches** directly mention the dragging action:
- "Drag to select/highlight..."
- "Using drag to highlight..."
- "Drag the mouse to select..."
- "Select by dragging..."

**Implicit Approaches** convey the dragging requirement without
mentioning "drag" or "dragging":
- Action-based: "Copy the sentence... "Highlight the two
paragraph... "Select to copy..."
- Range-based: "Select the range from... "Highlight across...
"Select all content between..."
- Span-based: "Select the text span... "Highlight the section
extending from..."

```
 - Multi-element: "Select all rows from X to Y "Highlight the
     multi-line text..."

\#\# Overall Guidelines

- Distribute expressions across both explicit and implicit
    approaches
- Ensure diversity of expressions across all categories
- For positional expressions, generate at least 3 expressions using
    relative positioning
- Each expression must clearly indicate that dragging is necessary.
    Expression should be unambuguious in terms of that 1) only the
    extracted target text span can match and all others within the
    screenshot cannot. 2) users are clear enough that they have to
    use drag to finish the goal.
- When generating the combination of referring expression and
    target text span (extracted from the screenshot via OCR), you
    should be as diverse as possible, i.e., you should find
    different target text spans from the screenshot. Thus, there
    shouldn't be duplication between the extracted target text span
    across different categories or even within one category.
If generating a referring expression that meets all requirements
    feels challenging, infeasible, or impossible for a category,
    return False for that category's availability.
- Last but not least, never omit any details of the target text
    span. You should output the full content of it.
```

## I.2 PROMPTS FOR GROUNDING ANNOTATION

```
You are an annotation assistant that grounds drag instructions to
    pixel coordinates.
Your input contains two components:
1. The target text span that must be selected via dragging. The
    text span preserves all punctuation and line breaks exactly as
    they appear in the screenshot.
2. The OCR parse of the screenshot represented as a JSON array of
    word-level entries. Each entry has the fields:
    - "id": an integer identifier that uniquely indexes the word in
    reading order.
    - "text": the word content exactly as recognized by OCR.
    - "bbox": the word bounding box given as [x\_min, y\_min,
    x\_max, y\_max] in pixels.
    - "confidence": the OCR confidence score between 0 and 1.

Task: determine which OCR word corresponds to the first token and
    which corresponds to the last token of the target text span.

Guidelines:
- Treat the OCR results as ordered by reading direction
    (left-to-right, then top-to-bottom). Use this order to resolve
    ties when multiple boxes contain the same text.
- Match the target span case-sensitively and include all
    punctuation, numbers, and special characters.
- Allow for minor OCR artifacts such as split words or stray
    spaces. If the span covers multiple consecutive OCR words,
    choose the id of the first word and the id of the last word in
    that contiguous sequence.
- Reject matches that require reordering or skipping words. The
    matching words must appear consecutively in the OCR stream.
- If the span appears multiple times, choose the occurrence whose
    surrounding context best matches the target text. Prefer the
    first perfect match when contexts are indistinguishable.

Reasoning procedure:
1. Normalize whitespace in both the target span and OCR tokens for
    comparison while keeping punctuation intact.
2. Scan the OCR sequence to locate candidate positions whose
    concatenated text matches the full span exactly.
3. Once a match is confirmed, record the id of the first word and
    the id of the last word in that sequence.
4. If no exact match is found, return that the span is not grounded
    and explain the issue.

Output format:
- Provide a short explanation of how the span was matched,
    mentioning any preprocessing that was required.
- Output a JSON object with the following keys:
    {
      "status": "grounded" | "not_grounded",
      "start_id": <integer or null>,
      "end_id": <integer or null>,
      "notes": <concise justification>
    }
- When "status" is "grounded", both ids must be integers and
    "notes" should summarize the matched text.
- When "status" is "not_grounded", set both ids to null and
    describe the failure condition in "notes".

Ensure the response is strictly valid JSON without additional
    commentary outside the JSON block.
```

## I.3 PROMPTS FOR FILTERING

```
You are an annotation validator who examines whether the annotated
    bounding boxes and referring expression jointly describe the
    same target text span.

You receive:
1. An annotated screenshot that contains either (a) one green
    bounding box and one red bounding box or (b) a single green
    bounding box.
2. A referring expression that describes the text span the user
    intends to drag-select.

Validation procedure:
- First reason carefully about the intended target text span
    implied by the referring expression. Explicitly restate the
    full span in plain text, including all punctuation required for
    a complete sentence when applicable.
- Verify that the referring expression itself is clear and
    unambiguous. If it cannot uniquely identify a span in the
    screenshot, mark the annotation as invalid.
- Simulate the drag selection:
  * When two boxes are provided, start the drag from the midpoint
    of the left edge of the green box and end at the midpoint of
    the right edge of the red box.
  * When only one box is provided, start and end the drag at the
    midpoints of the left and right edges of the green box,
    respectively.
- Compare the simulated drag span against the target text span
    inferred from the expression. The annotation is valid only if
    the simulated drag covers exactly the intended text (no missing
    or extra content).

Decision:
- If the annotation is valid and the expression is clear, output
    `is_valid: true`.
- Otherwise output `is_valid: false` and briefly explain the
    mismatch or ambiguity.
```

## J    ADOBE ACROBAT PDF READER EXAMPLE

Figure 10 shows an example of the Adobe Acrobat PDF Reader. There are a number of crucial features in the PDF Reader such as commenting, highlighting, that are only available after dragging the mouse to select the text.

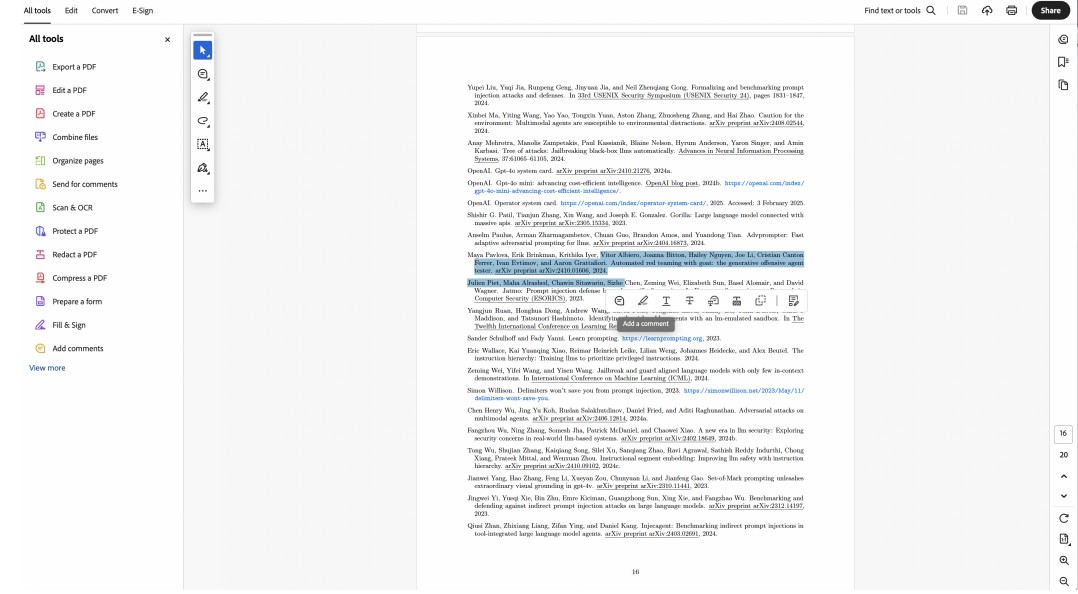

Figure 10: Adobe Acrobat PDF Reader Example

## K    OSWORLD EXAMPLES WITH TEXT SELECTION

After carefully examining the OSWorld examples, we find four examples that are related to text selection and list their task ids and instructions as follows:

- `72b810ef-4156-4d09-8f08-a0cf57e7cefe`
- I am peer-reviewing my friend's course outline. I think the last paragraph is redundant so I want to add strike-through on words in the last paragraph. Can you do this for me?

- `0810415c-bde4-4443-9047-d5f70165a697`
- Make the line spacing of first two paragraph into double line spacing

- `b21acd93-60fd-4127-8a43-2f5178f4a830`
- I have been practicing professional writing lately. Now I am writing essay which requires one paragraph each for introduction, body and conclusion with single-space for introduction, double-space for body then one-and-a-half-space for conclusion. The font size of this essay is 12. Could you help me on this?

Since we find that the planner model, i.e., o3, cannot properly trigger the drag action, we additionally add the instruction "You are encouraged to use the drag action to select the text whenever it is available" to original default the system prompt.

