# OpenReview forum: "Beyond Clicking: A Step Towards Generalist GUI Grounding via Text Dragging"
_ICLR.cc/2026/Conference — ICLR 2026 Conference Withdrawn Submission_

### Official Review · Reviewer_RWDf · 2025-10-31

**Soundness:** 3
**Presentation:** 2
**Contribution:** 3
**Rating:** 6
**Confidence:** 3

**Summary:**

This paper tackles text dragging for GUI agents, moving beyond the click-only focus of existing work. The authors create GUI-Drag and ScreenDrag benchmark with novel metrics, achieving 18% improvements over strong baselines including OpenAI CUA. The work is technically sound and addresses a real gap, revealing significant click-bias in current models. However, it's limited to document applications and relies quite heavily on OCR without error analysis.

**Strengths:**

S1: The paper convincingly argues why text dragging is essential for practical GUI agents, especially in productivity applications where text manipulation is central to user workflows.

S2: The ScreenDrag benchmark with three interface contexts (document, application, desktop) captures realistic deployment scenarios.

S3: GUI-Drag models achieve substantial improvements over strong baselines, including frontier models like OpenAI CUA.

**Weaknesses:**

W1: The benchmark only covers three document-centric applications (Word, PowerPoint, PDF readers). This limits generalizability to other GUI contexts where dragging is essential (e.g., drag-and-drop in file managers, graphic design tools, spreadsheets).

W2: The pipeline relies heavily on EasyOCR for word-level bounding boxes. OCR errors, particularly in dense text regions or with special formatting, may propagate through the system. The paper does not thoroughly analyze OCR failure modes or their impact on data quality.

W3: The case study includes only 3 tasks where human annotators agreed that text dragging is necessary. Though the review admits the difficulty of having more data for evaluation, this case is still insufficient to draw firm conclusions about real-world performance.

**Questions:**

Q1: How sensitive are the results to the threshold of $\phi$ being 3 in the SR metric? What happens with 2 or 5 pixels?

Q2: How do different OCR hyperparameters affect the results?

---

### Official Review · Reviewer_cqod · 2025-11-01

**Soundness:** 2
**Presentation:** 3
**Contribution:** 2
**Rating:** 2
**Confidence:** 4

**Summary:**

This paper addresses a critical gap in GUI grounding research: the field's over-reliance on 'click' actions, which neglects the essential 'drag' operation required for text selection in productivity applications. The authors introduce GUI-DRAG, a synthetic dataset for this task , and screendrag, a benchmark with three dedicated metrics. Using an continual training strategy , their resulting GUI-DRAG models substantially outperform existing baselines.

**Strengths:**

1. The work pinpoints a significant, practical limitation in existing GUI grounding models: their inability to perform text dragging , a core interaction in productivity software.
2. The paper presents a scalable, three-stage synthesis pipeline that overcomes the limitations of prior methods by using OCR and LLMs to generate a large-scale dataset.
3. The authors developed a new benchmark, screendrag, designed for evaluating this capability. It spans 3 levels of interface context and introduces three novel metrics that account for system-level behaviors like text snapping.

**Weaknesses:**

1. The paper's framing (e.g., the title "Beyond Clicking" and claims of moving towards a "generalist GUI grounding model") suggests a broad advancement. However, the work exclusively addresses text dragging. It seems to oversell its generality. It solves for "text dragging," not "dragging" as a general-purpose action in normal action spaces (e.g, the action space of os-atlas, arguvis, uitars). This should be framed as an important first step rather than a comprehensive solution.
2. The GUI-DRAG dataset, while large, raises some concerns. The instructions are generated by an LLM (GPT-4-mini) following predefined templates (e.g., positional, semantic, lexical). This raises questions about its (1) validity and (2) diversity. Real human instructions are often more context-dependent, and intent-driven rather than direct and explicit commands.
3. This threshold feels arbitrary. How sensitive are the final reported SR scores to this specific value?  Furthermore, the "text snapping" logic could be seen as an overly generous heuristic that rewards imprecise predictions.
4. Please refer to my questions.

**Questions:**

1. The paper does not quantify the error rate of the OCR itself or analyze how OCR failures (e.g., missed words, incorrectly merged lines, errors on non-standard fonts) propagate into the training data and evaluation. How robust is the model and the evaluation metric?
2. The paper identifies, but does not sufficiently investigate, the skill interference or catastrophic forgetting between click and drag tasks. Why does more click data negatively impact drag acquisition for the 7B model?

---

### Official Review · Reviewer_zHyG · 2025-11-03

**Soundness:** 2
**Presentation:** 2
**Contribution:** 3
**Rating:** 4
**Confidence:** 3

**Summary:**

This paper extends GUI grounding beyond traditional click actions to the underexplored yet essential text-dragging operation. The authors introduce GUI-DRAG (161K synthetic text-drag examples) and the corresponding benchmark (5,333 human-verified cases with three evaluation metrics). Using continual training on Jedi-3B/7B models, their GUI-DRAG-3B/7B significantly outperforms both open- and closed-source baselines (up to +18% absolute SR) while maintaining click performance. The work highlights the widespread “click bias” in current GUI agents and argues for more generalist grounding models.

**Strengths:**

1. Novel problem definition – one of the first to systematically explore text dragging in GUI grounding and providing concrete solutions.

2. Comprehensive dataset & benchmark – GUI-DRAG and SCREENDRAG provide scalable, diverse, and reproducible resources.

3. Rigorous evaluation – Three complementary metrics (DTR, B-Dist, SR) and detailed ablations across models, contexts, and granularities.

4. Practical contribution – Demonstrates effective continual training to integrate new actions without catastrophic forgetting.

**Weaknesses:**

1. The work mainly focuses on text-dragging tasks and has not been extended to other complex mouse-drag operations (e.g., box selection, sliding, or drag-and-drop of files). The evaluation is also limited to office applications and lacks data and tests for other scenarios such as web pages, code editors, and other applications.
2. The OSWorld case study is limited: an improvement of only +1 is not statistically significant. The authors should either increase the number of evaluation tasks or replace this section with the material in Appendix H to extend discussion on different subsets.

**Questions:**

1. Why do Jedi-7B and GUI-DRAG-7B perform worse on SCREENGRAG than their 3B variants?
2. In the continued-pretraining analysis, why was JEDI data chosen? The model may already have been exposed to these examples, so further pretraining could simply upweight them. Why not continue pretraining on data sampled from a different distribution?

---

### Official Review · Reviewer_YEE9 · 2025-11-08

**Soundness:** 2
**Presentation:** 2
**Contribution:** 3
**Rating:** 4
**Confidence:** 4

**Summary:**

This paper focuses on a novel perspective of GUI Agents, text tragging. The main contribution is a synthesized dataset, named GUIDrag as well as the the corresponding benchmark ScreenDrag. The models continually-trained on GUIDrag achieve substantial improvements while preserve the clicking performances.

**Strengths:**

1. The topic on text dragging is promising and interesting.
2. This paper offers a dataset GUIDrag and a benchmark ScreenDrag.
3. The performances and findings are meaningful.

**Weaknesses:**

1. It is unclear how often do we need to use text dragging in real-life scenarios (e.g., OS-World benchmark) ? The authors should provide some statistics to support the importance. Can text dragging be replaced by other compositional actions?

2. The coverage of the benchmark. Based on the illustrated cases, this benchmark only focuses on doc-like cases. But, more cases in excel, powerpoint and social media app are also important. Does ScreenDrag cover these scenarios? And does ScreenDrag cover diverse platforms (e.g., Web / Mobile / Desktop) ?

**Questions:**

See above.

---

### Note · Authors · 2025-11-30

I have read and agree with the venue's withdrawal policy on behalf of myself and my co-authors.